# Experimental Animal Studies Support the Role of Dietary Advanced Glycation End Products in Health and Disease

**DOI:** 10.3390/nu13103467

**Published:** 2021-09-29

**Authors:** Melpomeni Peppa, Ioanna Mavroeidi

**Affiliations:** Endocrine Unit, 2nd Department of Internal Medicine-Propaedeutic, Attikon University Hospital, National and Kapodistrian Athens University Medical School, 12462 Athens, Greece; joannamavroeidi@gmail.com

**Keywords:** advanced glycation end products, dietary advanced glycation end products, dietary glycotoxins, oxidative stress, diabetes mellitus, obesity, cardiovascular disease

## Abstract

The increased incidence of obesity, diabetes mellitus, aging, and associated comorbidities indicates the interplay between genetic and environmental influences. Several dietary components have been identified to play a role in the pathogenesis of the so-called “modern diseases”, and their complications including advanced glycation end products (AGEs), which are generated during the food preparation and processing. Diet-derived advanced glycation end products (dAGEs) can be absorbed in the gastrointestinal system and contribute to the total body AGEs’ homeostasis, partially excreted in the urine, while a significant amount accumulates to various tissues. Various in vitro, in vivo, and clinical studies support that dAGEs play an important role in health and disease, in a similar way to those endogenously formed. Animal studies using wild type, as well as experimental, animal models have shown that dAGEs contribute significantly to the pathogenesis of various diseases and their complications, and are involved in the changes related to the aging process. In addition, they support that dAGEs’ restriction reduces insulin resistance, oxidative stress, and inflammation; restores immune alterations; and prevents or delays the progression of aging, obesity, diabetes mellitus, and their complications. These data can be extrapolated in humans and strongly support that dAGEs’ restriction should be considered as an alternative therapeutic intervention.

## 1. Introduction

The increased incidence of various “modern” diseases such as obesity, insulin resistance, diabetes mellitus, and their complications, as well as the normal aging process and its related morbidities, indicates the interplay between genetic and environmental influences.

A large body of evidence supports that advanced glycation end products (AGEs) are involved in the pathogenesis of those diseases and their complications [1,2,3,4,5]. In addition, it is now largely supported that diet-derived AGEs (dAGEs) play an important role as well [1,2,3,4,5].

Methods of food preparation and processing have a significant effect on the generation of diverse unstable carbonyl derivatives of glyco- and lipoxidation reactions, which can be absorbed in the gastrointestinal system and partially excreted in urine [1,2,3,4,5]. The socioeconomic changes in the past 50 years and new technologies employed in the mass production of food, including heat, dehydration, ionization, irradiation, and flavor, have led to the production of various toxic substances in modern foodstuffs, including dAGEs [6,7,8,9].

The increased consumption of dAGEs contributes to the total AGEs’ body burden, in a similar way to those endogenously formed, and leads to suppression of host defenses and induction of intracellular reactive oxygen species, which can shift basal oxidative stress and lead to inflammation, insulin resistance, obesity, β-cell dysfunction, impaired insulin secretion and insulin resistance, and diabetes, as well as their comorbidities, and mediate the aging process (Figure 1).

Studies using wild type, as well as experimental, animal models have been very important to clarify the role of dAGEs in various pathologies. These studies emphasize that dAGEs’ restriction reduces basal oxidative stress and inflammation; restores host defenses and increases life span; and prevents or improves dementia, type 1 and type 2 diabetes mellitus, renal disease, and their complications [3].

In this chapter, we will review data emerging from animal studies supporting the role of dAGEs in health and disease.

## 2. Dietary AGEs’ Homeostasis—Animal Studies

Kinetic studies in normal rats showed that about 10% of dAGEs are absorbed via the gastrointestinal tract. The major "exit" of dAGEs is the kidney, by filtration or by active uptake and secretion, two processes that result in the net excretion of AGEs in the urine [10,11]. Increased dAGEs’ uptake or decreased renal AGEs’ clearance can potentially lead to accumulation of prerenal AGEs and oxidative stress in both renal and extrarenal tissues [10,11]. It has been estimated that two-thirds of dAGEs remain in contact with tissues for 72 h, and that this rate is greater than that of those that are endogenously formed. The actual effect of sustained exposure to dAGEs was evaluated in a series of studies in mice. As described later, several animal studies provide a new framework and an experimental setting from which to revisit the pathogenesis and treatment of various pathologies.

Animal studies using wild type, as well as different experimental, animals of diabetes, obesity, insulin resistance, aging, and anti-AGEs agents (Table 1) showed that dAGEs are absorbed, contribute to the total AGEs’ homeostasis, and are implicated in the pathogenesis of various diseases and their complications, in a similar way to their endogenously formed counterparts. The existing studies have used parenteral administration of AGEs mixtures or high AGEs (H-AGEs) and low AGEs (L-AGEs) diets. Both diets were nutritionally equivalent and were pelleted by the manufacturer and kept at 4 °C. The H-AGEs diet was exposed to an autoclaving cycle (exposure to vacuum, autoclaving, drying), as per standard procedure, and fortified with supplements to offset heat-depleted micronutrients. The L-AGEs diet was prepared without exposure to heat. The diets differed in the AGEs content, which was 3- to 5-fold lower in the L-AGEs diet, based on direct assessment of AGEs by ELISA, using specific monoclonal antibodies. The data from animal studies were further confirmed by clinical studies [1,2,3,4,5,12] using diets with different amounts of AGEs according to a large database of different foods, which were subjected to standard cooking methods (boiling, broiling, deep-frying, oven-frying, and roasting) [13,14].

A significantly reduced intake of dAGEs was achieved by increasing the consumption of fish, legumes, low-fat milk products, vegetables, fruits, and whole grains and by reducing intake of solid fats, fatty meats, full-fat dairy products, and highly processed foods. Based on this database, a number of clinical studies have used H-AGEs diet (>15,000 kU AGEs) and L-AGEs diet (<15,000 kU AGEs) in subjects with different pathologies, such as insulin resistance, diabetes mellitus, chronic kidney disease, aging, and obesity, showing that decreased dAGEs intake improves the diseases and their complications. [13,14].

## 3. Dietary AGEs’ Effects on Health

Intraperitoneal injections of AGEs in healthy male Sprague–Dawley rats for 16 weeks resulted in impaired insulin secretion, decreased islet preproinsulin gene expression, and a characteristic decline in first-phase insulin secretion in response to glucose. These changes were attributed to the stimulation of the superoxide production within mitochondria and the interruption of several critical sequential steps, including changes in cellular glucose uptake, alteration of cellular calcium flux, and loss of mitochondrial ATP content, prevented by alagebrium [22,39].

In addition, parenteral or oral administration of AGEs in healthy mice leads to a cycle of increased cell and tissue AGEs, oxidative stress, inflammation, and tissue injury similar to the diabetic vascular or renal complications, without hyperglycemia, reversed by anti-AGE agents [31,40,41,42,43,44]. These studies indicate that dAGEs contribute significantly to the total body AGEs burden and induce oxidant stress, impaired insulin secretion, insulin resistance, and inflammation in healthy mice [3,38]. They also demonstrate that these changes can be diminished, and further pathology can be prevented in several mouse models by restricting the intake of dAGEs without reducing energy intake [29,32,33,34,38,45].

## 4. Dietary AGEs’ Effects in Diabetes Mellitus 

### 4.1. Type 1 Diabetes Mellitus (T1D)

A large body of evidence in experimental animals supports that AGEs and dAGEs are involved in the pathogenesis of T1D, either by a direct effect on the β-cell, causing defective insulin secretion and cell apoptosis, or by induction of systemic inflammation, oxidative stress, and alterations in the immune cell activation [22,29,34,39,46].

Peppa et al. showed that early exposure of NOD mice to increased dAGEs had a role in the pathogenesis of T1D. The mice were exposed to a high-AGEs diet (H-AGEs) and to a nutritionally similar diet with approximately fivefold lower levels of N^ε^-carboxymethyllysine and methylglyoxal-derivatives (L-AGEs), owing to the shorter heat exposure during processing. The suppressive effects of diabetes under L-AGEs feeding were significant across three generations in founder mice, initiated at 3 or 6 weeks of age, and were extended throughout two generations of offspring, F1 and F2, if kept on the L-AGEs maternal regimen. L-AGEs fed mice exhibited a diabetes-free rate of 86%, greatly increased survival rates by 76% for up to 56 weeks of age, and displayed a delay in disease onset (4-month lag). Survival for L-AGEs fed mice was 76 versus 0% after 44 weeks of H-AGEs fed mice. T1D reduction (33 to 14%) coincided with initiation to the L-AGEs diet during the perinatal period. The greatest disease prevention (14%) was associated with exposure to a low AGEs maternal environment, suggesting that toxic AGEs are transportable via the placenta. However, the protective maternal effect of the L-AGEs diet was reversed after crossing over to the H-AGEs diet, as readily as the normally H-AGEs maternal effect was reversed by the L-AGE diet, when applied at weaning (3 or 6 weeks). These data confirmed the plasticity inherent in this period and reinforced the importance of “dose”, “time”, and “duration” of exposure to toxic factor(s). The majority of mice exposed to the L-AGEs environment maternally or at weaning displayed modest insulitis and no diabetes for 1 year. A milder insulitis and a milder diabetes characterized L-AGEs fed mice that did become diabetic, as compared with those with severely damaged islets and overt diabetes seen in H-AGEs fed mice by 25 weeks [29]. Furthermore, the 16-week-old prediabetic L-AGEs fed NOD mice exhibited near-normal glucose and insulin responses to glucose challenge compared with the typically dysfunctional pattern of age-matched H-AGEs fed mice. Reduced insulitis in L-AGEs versus H-AGEs fed mice was marked by GAD- and insulin-unresponsive pancreatic interleukin (IL)-4-positive CD4+ cells compared with the GAD and insulin-responsive interferon (IFN)-γ positive T cells from H-AGEs fed mice. Spleenocytes from L-AGEs fed mice consisted of GAD- and insulin-responsive IL-10-positive CD4+ cells compared with the IFN-γ positive T-cells from H-AGEs fed mice [29]. A marked suppression of total pancreatic lymphocytes was observed in the L-AGEs fed mice (15-fold) compared with the H-AGEs NOD group fed a regular diet. In addition, the L-AGEs fed mice displayed a pattern of predominantly IL-4-positive CD4+ cells. There were virtually no infiltrates in islets from two generations of mice exposed to L-AGEs nutrients during the fetal stage or early in life. In these mice, there was also a marked unresponsiveness of pancreatic T-cells to β-cell antigens, insulin, and GAD. This might indicate blocked autoreactive T-cell recruitment owing to low local expression of these antigens under a “safer” AGEs-poor diet [29].

Borg et al. studied parental male and female NOD mice that followed a low or high in AGEs diet maintained throughout gestation and lactation. From conception to early postnatal life, an L-AGEs diet decreased circulating AGEs’ concentration in female offspring compared with an H-AGEs diet [24]. Insulin gene, glucagon, AGEs, and the AGEs receptor-1 (AGER-1) expression were greater in islet cells isolated from offspring in the L-AGEs diet group than in age-matched nondiabetic control mice. Insulin, proinsulin, and glucagon secretion were greater and infiltration of islet immune cells was lower in the L-AGEs group [24].

In another study, Borg et al. showed that only one-third of NOD mice on an L-AGEs diet developed T1D. In addition, the incidence of T1D decreased to less than 15% over the next two generations. The AGEs-restrictive NOD mice had minimal insulitis, while the NOD mice fed an H-AGEs diet had the typical hyperresponsive, interferon-γ-positive, interleukin-4-negative islet T cells that do not respond to common islet antigens such as glutamic acid decarboxylase or insulin [22,24].

Therefore, high AGEs intake may provide excess antigenic stimulus for T-cell-mediated diabetes or direct β-cell injury in NOD mice being ameliorated by maternal or neonatal exposure to L-AGEs nutrition. The above data indicate that dAGEs are involved in the pathogenesis of T1D [29]. Dietary AGEs restriction results in suppression of islet infiltration by β-cell cytotoxic T-cells and islet toxicity, significantly delayed onset, reduced severity of disease, and marked increase in overall survival [29,38]. However, future studies are required to define how exposure to excess AGEs specifically contributes to T1D development and autoimmunity, and whether this is via direct or indirect pathways. Taken together, the existing data suggest dAGEs as a modifiable risk factor and therapeutic target for the prevention of T1D.

### 4.2. Type 2 Diabetes Mellitus (T2D)

AGEs have been linked to the pathogenesis of insulin resistance (IR), which is the main underlying mechanism of type 2 diabetes (T2D) [47]. A large body of experimental evidence shows that dAGEs are also involved in the development of IR.

Hofmann et al. showed that already obese and diabetic C57/BL/KsJ db/db mice fed with L-AGEs diet ceased to gain weight within 10 weeks of study and, by 20 weeks, they exhibited a significant weight loss and better glycemic profile. L-AGEs fed mice exhibited lower fasting insulin levels, a nearly complete restoration of glycemic response to intravenous glucose, and insulin tolerance tests, indicating a well-preserved pancreatic islet morphology and function [33,48]. L-AGEs fed mice showed an improved glucose uptake in abdominal adipose tissue and increased HDL cholesterol levels [33]. This study supported that dAGEs contribute to the pathogenesis of IR and dietary AGE restriction seems to be effective in the prevention of this process [33].

Sandu et al. showed that, after 6 months on high-fat (HF) and H-AGEs (H-AGEs-HF) diet, 75% of normal C57/BL6 mice were diabetic and exhibited higher body weight, fasting glucose, insulin, serum AGEs, impaired glucose and insulin responses during glucose tolerance tests, euglycemic and hyperglycemic clamps, altered pancreatic islet structure and function, higher plasma 8-isoprostanes, and lower adiponectin levels, compared with those on L-AGEs-HF diet. This group also showed more visceral fat (by two- and fourfold) and more AGEs modified fat (by two- and fivefold) than the L-AGEs-HF and control mice, respectively, despite similar dietary fat intake and body weight gain by both groups. None of the mice on the L-AGEs-HF diet were diabetic, despite a similar rise in body weight and plasma lipids. This study emphasizes the protective role for a low-AGEs diet, even if the fat content is elevated [34].

Cai et al. performed a study in four generations of F3 mice that exhibit increased adiposity and premature IR, marked by severe deficiency of AGEs receptor 1 (AGER-1) and of survival factor sirtuin 1 in white adipose tissue, skeletal muscle, and liver. The mice were fed isocaloric diets either with or without AGEs (H-AGEs and L-AGEs, respectively) [35]. F3-H-AGEs mice exhibited increased adiposity, glucose intolerance, hyperinsulinemia, elevated leptin, and decreased adiponectin levels, and appeared prematurely, namely 6 months before its onset in control mice and 12 months before onset in F3-LAGEs mice. They also showed increased lipids, enriched with AGEs, expanded adipose tissue, a marked increase in NF-κB acetylated-p65 in adipocytes, and high TNF-α and CD11c in peritoneal macrophages, as well as suppressed AGES receptor 1, survival factor sirtuin 1, PPARγ, IL-10, and CD26 mRNA levels, compared with the control mice. This study indicates that the increased dAGEs produced in thermally treated food can profoundly alter the inflammation/insulin axis [35].

Mastrocola et al. have shown that diets high in AGEs promote liver AGEs’ accumulation, affecting liver enzymes’ production and action, leading to high plasma levels of the sphingolipid intermediates (e.g., ceramide and sphingosine-1-phosphate) and IR [18]. This was further supported by the beneficial effects of pyridoxamine supplementation, which diminished hepatic AGEs and prevented alterations of sphingolipid metabolism and the development of IR in H-AGEs fed FD mice [18].

Verboven et al. showed that Sprague–Dawley rats fed with a “Western” diet for 18 weeks exhibited increased AGEs levels, dyslipidemia, and hyperglycemia in combination with altered heart function, suggesting that a “Western” diet that is rich in AGEs might be a cause of DM and cardiovascular disease, which often present together [19].

### 4.3. Diabetic Complications

#### 4.3.1. Cardiovascular Disease 

Several studies in experimental animals have shown that dAGEs play an important role in the pathogenesis of micro- and macroangiopathy, in a similar way to those endogenously formed.

Lin et al. showed that genetically hypercholesterolemic apolipoprotein E-deficient (apoE−/−); streptozotocin-induced diabetic mice fed with an L-AGEs diet exhibited lower AGEs levels; 50% smaller lesions at the aortic root; markedly suppressed tissue AGEs, AGE-R- 1, AGER- 2, and AGER expression; and reduced numbers of inflammatory cells, tissue factor, vascular cell adhesion molecule-1, and MCP-1 levels. This study indicated an important link between dAGEs, tissue AGEs’ accumulation, and diabetes-related vascular disease, and supported a beneficial role of dAGEs’ restriction [49]. Furthermore, Lin et al. showed that apoE−/−, streptozotocin-induced diabetic mice, on an L-AGEs diet, showed a significant decrease in circulating AGEs levels, intimal formation, neointimal area, intima/media ratio, and stenotic luminal area, at 4 weeks after femoral artery injury. A marked reduction (56%) in macrophages in the neointimal lesions, as well as an obvious reduction in smooth muscle cell content, and a reduced deposition of AGEs in the endothelia, SMC, and macrophages in neointimal lesions, were also observed [50].

Deng et al. investigated male apoE−/− mice, divided into four groups based on the feeding periods (0, 2, 4, and 6 months), and further divided them into three subgroups corresponding to the following diet treatments: regular diet, high-fat diet + normal saline injection, and high-fat diet + aminoguanidine injection [17]. A high fat diet, meaning high AGEs intake, was associated with increased total and low-density cholesterol levels, transaortic valve velocity, calcification, and AGEs’ deposition around the aortic valve surfaces, mediated through AGEs’ interaction with AGER-1 and the counteraction of BMPR2 and TGF-β1 signaling. These changes were alleviated and attenuated with the AGE inhibitor aminoguanidine, which inhibited the AGEs’ interaction with AGER-1 [17].

Haesen et al. showed that adult male Sprague–Dawley rats, under daily injections with high molecular weight AGEs, for six weeks, showed intracardiac pressure overload, characterized by increased systolic and mean pressures, increased contraction response in aortic rings to phenylephrine, impaired relaxation in response to acetylcholine, increased collagen deposition, and intima-media thickness of the aortic vessel wall, compared with the control mice group [51]. This study showed an important contribution of high molecular weight AGEs, which are present in the Western diet, in the development of cardiovascular disease. 

Peppa et al. showed that female db/db mice, after 3 months on an L-AGEs diet, displayed lower circulating and skin AGEs deposits, increased epithelialization, angiogenesis, inflammation, granulation tissue deposition, enhanced collagen organization, and more rapid wound closure time, compared with H-AGEs fed mice. This study suggests that dAGEs’ restriction might improve impaired diabetic wound healing [30,52].

Verboven et al. showed that Sprague–Dawley rats fed a “Western” diet for 18 weeks exhibited increased AGEs levels and altered heart function with increased end-diastolic pressure and left ventricle hypertrophy, in addition to hyperglycemia and dyslipidemia. This study highlights the importance of the “Western” diet, a diet rich in AGEs, as a cause of cardiovascular disease and DM [19].

#### 4.3.2. Diabetic Nephropathy

Many studies support the role of AGEs in diabetic nephropathy development and progression to end stage renal disease. Among them, animal studies give some impressive results for dAGEs.

L- AGEs fed NOD mice showed lower serum and renal AGEs, a minimal glomerular pathology, a modest increase in urine albumin/creatinine ratio, and significantly prolonged survival, compared with the H-AGEs fed mice [38].

Yuan et al. showed that intraperitoneal injections of a high-fat, high-sugar diet in streptozocin induced male Sprague-Dawley diabetic rats resulted in strong staining of Oil Red O in the renal tubules, increased serum and renal AGEs’ deposition, upregulation of both mRNA and protein expression of 3-hydroxy-3-methylglutaryl coenzyme A reductase, LDL receptor, sterol regulatory element binding protein-2, and decreased urine protein and u-NGAL. It is noteworthy that these changes were reversed or prevented by the simultaneous administration of aminoguanidine and anti-AGER agents [23]. This study indicates, in addition to the harmful effects of an H-AGEs diet, that AGEs induce DN via disruption of intracellular feedback regulation of cholesterol, while an anti-AGEs strategy decreases lipid accumulation, offering a renoprotective role in the progression of DN [23].

## 5. Dietary AGEs’ Effects on Obesity

AGEs are involved in the pathogenesis of obesity, and dAGEs seem to also be involved in a similar manner.

Rajan et al., using a proteome-cytokine array kit, evaluated 40 circulatory proinflammatory cytokines and chemokines in experimental Swiss albino mice fed with H-AGEs and regular diet in the presence and absence of curcumin and gallic acid for 6 months. He showed that H-AGEs fed mice displayed increased body weight. In addition, five cytokines (IL-16, IL-1α, ICAM, TIMP-1, and C5a) were found to be highly expressed (3.5-fold), eleven cytokines were moderately expressed (2-fold), three chemokines (BLC, SDF-1 and MCP-1) were highly expressed (4-fold), and ten showed moderate expression (2-fold), in H-AGEs fed mice compared with control mice. These effects were prevented by the co-treatment with curcumin and gallic acid. This study indicates that dAGEs induce obesity and systemic inflammation, which are rather parallel phenomena [16].

Sandu et al. showed that, after 6 months on an HF and H-AGEs (H-AGEs-HF) diet, 75% of normal C57/BL6 mice exhibited higher body weight, in addition to changes in glucose homeostasis, compared with an HF and L-AGEs diet, indicating the weight promoting effect of increased dAGEs’ consumption [34].

Hofmann et al. showed that already obese and diabetic C57/BL/KsJ db/db mice fed an L-AGEs diet ceased to gain weight within 10 weeks of study and, by 20 weeks, they exhibited a significant weight loss and better glycemic profile, compared with the H-AGEs fed mice. This study supported that dAGEs contribute to the pathogenesis of obesity in combination with other components of the metabolic syndrome [33].

## 6. Dietary AGEs Effects on Aging

Studies have shown that AGEs are involved in the aging process and its complications, and the same seems to be the case for dAGES. 

Cai et al. showed that C57BL6 male mice on a lifelong exposure to an L-AGEs diet exhibited higher than baseline tissue AGER-1 levels and glutathione/oxidized glutathione ratio, as well as decreased plasma 8-isoprostanes, tissue AGER-1, and p66shc levels, compared with the control group. This was associated with decreased systemic AGEs’ accumulation, amelioration of IR, albuminuria, glomerulosclerosis, and extended lifespan. This study showed that dAGEs’ restriction may preserve from life-long exposure to high levels of glycoxidants that exceed AGER-1 and antioxidant reserve capacity, resulting in decreased tissue damage and a longer lifespan [38].

In another study, Cai et al. showed that pair-fed mice on a calorie-restricted H-AGEs diet developed high levels of 8-isoprostanes, AGEs, AGER-1, and p66shc, coupled with low AGER-1 and glutathione/oxidized glutathione ratio, IR, marked myocardial and renal fibrosis, and shortened lifespan, compared with those on a calorie-restricted diet alone. This study showed that H-AGEs intake competed against the benefits of calorie-restriction and that healthspan seems to be increased by calorie-restriction in terms of consumption of foods containing L-AGEs [36].

## 7. Dietary AGEs’ Effects on Dementia 

Dementia has been associated with increased oxidant stress and AGEs [53,54,55], while dAGEs seem to play an important role as well.

Cai et al. studied MG+/MG− mice exposed throughout life to an L-AGEs diet (MG-), a supplemented AGEs diet (MG+), and regular chow. The MG+ fed mice developed metabolic syndrome, increased brain amyloid-β42 and AGEs’ deposits, gliosis, and cognitive deficits, accompanied by suppressed SIRT1, nicotinamide phosphoribosyl transferase, AGER-1, and PPARγ, changes that seem to be prevented in L-AGEs fed mice. This study indicates the importance of dAGEs’ restriction in the development and progression of dementia [53].

Huang et al., using healthy male mice on different diets (a typical diet and free drinking, a standard diet and 300 mg/L lead acetate solution, and a high-fat diet and free drinking) for 10 weeks, showed that the high-fat fed mice exhibited lipid metabolism disorders and impaired cognitive function and central nervous system. This was mediated by promoting the secretion of inflammatory factors in glial cells, inducing the inflammatory reaction of brain tissue, inhibiting glutathione antioxidant status, and increasing the AGEs’ content, which further aggravates the injury of brain tissue [25].

Maciejczyk et al. studied male Wistar rats on high and standard protein diets for 8 weeks. The mice on a high protein diet showed higher uric acid concentration and activity of the glutathione hyperoxidase system, catalase, and lower glutathione levels in the hypothalamus, whereas in the cerebral cortex, these parameters remained unchanged. Both brain structures expressed a higher content of 4-hydroxynonenal and malondialdehyde and increased AGEs in the hypothalamus. The observed differences among the studied brain compartments suggest that the hypothalamus is more susceptible to oxidant stress caused by a high protein diet rich in AGEs [26].

## 8. Dietary AGEs’ Effects on Muscle and Bone

Diets enriched with AGEs have recently been related to muscle and bone dysfunction processes [56].

Egawa et al. showed that mice on an H-AGEs diet for 16 weeks exhibited higher levels of Nε -(carboxymethyl)-L-lysine, a marker for AGEs, in extensor digitorum longus muscles, as well as lower muscle strength and endurance and decreased mRNA expression levels of myogenic factors, including myogenic factor 5 and myogenic differentiation, in extensor digitorum longus muscle, compared with L-AGEs fed mice. This study indicates that long-term exposure to an increased dAGEs’ consumption disrupts the skeletal muscle growth and contractile muscle function owing to the inhibition of myogenic potential and protein synthesis [56].

IIIien-Junger et al. studied female and male mice on L-AGEs and H-AGEs diets, for 6 and 18 months. The H-AGEs fed female mice exhibited increased AGEs levels in serum and cortical vertebrae, an inferior vertebral trabecular structure with decreased bone mineral density and bone volume fraction, functional deterioration with reduced shear and compression moduli, and maximum load to failure after 6 months. At 18 months, H-AGEs fed mice of both sexes had significant, but small decreases in cortical BMD and thickness, and functional biomechanical behaviors compatible with those observed in aging mice. This study showed that increased dAGEs’ consumption without morbidities such as diabetes or obesity have an adverse effect on vertebral microstructure, mechanical behaviors, and fracture resistance in young female mice, in a manner suggesting accelerated bone aging [28].

## 9. Dietary AGEs’ Effects on the Gut

Qu et al. showed that Sprague-Dawley rats on an H-AGEs diet for 6, 12, or 18 weeks exhibited a reduction in the diversity and richness of the microbiota, especially saccharolytic bacteria such as Ruminococcaceae and Alloprevotella, which can produce small chain fatty acids (SCFAs), whereas some putatively harmful bacteria (Desulfovibrio and Bacteroides) were increased [27].

Mao et al. studied hyperlipidemic rats fed diets consisting of fish protein, 6% low level glycated fish protein, and 12% high level glycated fish protein for 4 weeks. The mice fed with the low-level glycated fish protein displayed significantly changed protein fermentation and reduced inflammation markers and blood lipids, but increased AGEs’ plasma accumulation and fecal excretion and butyrate producing Ruminococcus_1 and Roseburia [21]. Furthermore, the high level of glycated fish protein diet significantly decreased Ruminiclostridium_6, Desulfovibrio, Helicobacter, and Lachnospiraceae groups. These changes reflect the effect of dAGEs on the the modulation of gut microbiota composition and fermentation metabolite profiles, emphasizing the beneficial effect of L-AGEs on gut health [21,57]. 

## 10. Dietary AGEs’ Effects on Reproductive Function 

AGEs have been involved in various parts of reproductive function in both males or females, and dAGEs seem to play an important role as well.

Thornton et al. showed that mice fed an H-AGEs diet spent a significantly longer time in the diestrus phase, had a similar number of oocytes released following ovarian superovulation, showed increasing mRNA expression in genes involved in steroidogenesis (Star) and folliculogenesis, showed increased expression of macrophage markers in the ovary, and had significantly fewer corpora lutea, compared with the L-AGEs group [15]. This study showed that increased dAGEs’ consumption impacts fertility and, in general, ovarian function, regardless of obesity or other morbidities [15].

Kandaraki et al. studied normal nonandrogenized and androgenized prepubertal rats, fed H-AGEs and L-AGEs diets. She showed that H-AGEs fed mice had lower ovarian glyoxalase-1 activity, compared with L-AGEs and control groups. In addition, ovarian glyoxalase-1 activity was positively correlated with body weight gain as well as estradiol and progesterone levels, and negatively correlated with AGEs’ expression in the ovarian granulosa cell layer of all groups [37]. This study indicated that ovarian tissue seems to be a new target of AGEs’ deposition, leading to hyperandrogenic disorders such as PCOS.

Janšáková et al. studied pregnant rats fed a standard or an AGEs-rich diet from gestation day 1, while a third group received a standard diet and AGEs (CML) via gavage. Rats fed an H-AGEs diet exhibited higher plasma AGEs levels. However, this study does not support the speculation that dAGEs might contribute to the pathogenesis of pregnancy complications, and it cannot be speculated whether this is the cause or result, but the short duration of the rodent gestation warrants further studies analyzing the long-term effects of AGEs in preconception nutrition [58].

Regarding testicular function, increased dAGEs’ intake resulted in increased serum AGEs and histopathological damage in the testes and epididymides, a decreased total number of epididymal sperm, and an increased abnormal sperm rate AGER-1 and malondialdehyde levels, compared with the control group. These changes were restored after administration of silymarin, a natural AGEs inhibitor, further emphasizing the role of dAGEs in testicular health [20].

Akbarian et al. studied twenty-five 4-week-old C57BL/6 mice, fed diets differing in fat content (control, 45% high fat, 60% high fat, 45% high fat/AGEs, 60% high fat/AGEs). After 18 months, sperm motility, DNA fragmentation, and protamine deficiency, as well as membrane and cytoplasmic peroxidation, were negatively affected by the high fat and high fat/AGEs diets, in addition to changes in glucose and insulin homeostasis [59].

## 11. Dietary AGEs Effects on Cancer

AGEs have been associated with tumor development either directly through their increased formation because of increased glucose uptake and glycolysis or indirectly through insulin resistance, oxidative stress, and chronic inflammation induction.

Van Heijst et al. studied the presence of AGEs in human cancer tissues and detected the presence of the AGEs Nε-(carboxymethyl) lysine) (CML) and argpyrimidine in several human tumors using specific antibodies, suggesting that AGEs could be implicated in the biology of human cancer [60]. Jiao et al., in a study of 2193 pancreatic cancer cases (1407 men and 786 women) with a 10.5-year follow-up, found that men in the highest quintile of red meat consumption had a higher risk of pancreatic cancer (HR: 1.35; 95% CI: 1.07, 1.70), which attenuated after adjustment for CML-AGE consumption (HR: 1.20; 95% CI: 0.95, 1.53) [22]. This study suggested that consumption of meat that is enriched in Nε-(carboxymethyllysine)-AGE might contribute to pancreatic cancer through mechanisms other than meat mutagens, such as increased AGEs’ formation [61]. There a potential mechanistic link reported between carbohydrate-derived metabolites and cancer, which may provide a biologic consequence of lifestyle that can directly affect tumor biology. Failure due to AGEs can lead to protein damage, aberrant cell signaling, increased stress responses, and decreased genetic fidelity [62].

## 12. Limitations of Animal Studies in dAGEs

Animal studies have many limitations in general. In particular, the animal studies that have investigated dAGEs’ effects in health and disease present a significant heterogenicity in study design, dietary patterns related to the AGEs’ content, AGEs’ measurements, and the presentation of results. In addition, the existing studies estimate some of the many AGEs compounds, although εN-carboxymethyl-lysine and methylglyoxal are good representatives, as they are increased in various pathologies and associated with markers of health and disease. The short follow-up duration is considered to be a problem, as the adverse consequences of an H-AGEs diet may arise over a period of years rather than weeks or months. However, the fact that H-AGEs and L-AGEs diets were associated with markers of disease in various experimental animal models support their role in various pathologies.

Furthermore, the existing food AGEs database also has limitations, including the fact that foods were selected from diets common in the U.S. population, and thus may not represent the international average. Another limitation is that only two of many AGEs were measured. However, the fact that both diets were associated with markers of disease in healthy subjects and were elevated in patients with diabetes and kidney disease lends credibility to their role as pathogens in foods consumed by the public and persons with certain chronic diseases.

There are also conflicting reports that present evidence against an association between intake of dAGEs and their presence in the body or their real pathological effects [63,64].

There are also studies supporting that dAGEs have no systemic effects. Semba et al. found that a high- or low-AGEs diet had no significant impact on peripheral arterial tonometry or any inflammatory mediators after 6 weeks of dietary intervention [65]. Kellow et al. concluded that there is insufficient evidence to recommend dietary AGEs’ restriction for the alleviation of the proinflammatory milieu in healthy individuals and patients with diabetes or renal failure, limiting the conclusions that can be drawn [66].

Future studies should continue to expand the dAGEs database and investigate additional methods for reducing AGEs’ generation during home cooking and food processing. 

## 13. Conclusions

A large body of evidence from experimental in vivo, in vitro, as well as clinical studies have indicated the important role of AGEs in the pathogenesis of various pathologies and their complications. Animal studies, as well as in vitro and clinical data, further support the role of dAGEs in health and disease. The animal studies that have investigated the dAGEs’ effects on health and disease, besides the general limitations of this kind of study, presented a significant heterogenicity in study design, AGEs’ assessment, and the presentation of results, which makes it difficult to perform a metanalysis. Dietary AGEs intake was poorly reported in several articles. Moreover, it is likely that there may have been differences other than the AGEs’ content between interventions. Finally, the included studies had a relatively short follow-up duration. The adverse consequences of an H-AGEs diet may arise over a period of years rather than weeks or months, as AGEs accumulate, glycate other proteins, or are incorporated into tissues. Therefore, studies of longer duration are required. However, the fact that H-AGEs and L-AGEs diets were associated with markers of disease in various experimental models support their role in various pathologies.

In addition, the existing food database also has limitations, including the fact that foods were selected from diets common in the U.S. population, and thus may not represent the international average. Another limitation is that only two of many AGEs were measured. However, the fact that both diets were associated with markers of disease in healthy subjects and were elevated in patients with diabetes and kidney disease lends credibility to their role as pathogens in foods consumed by the public and persons with certain chronic diseases. However, ongoing studies are needed to further expand the dAGEs database and investigate additional methods for reducing AGEs’ generation during home cooking and food processing. Future studies should continue to investigate the health effects of AGEs and refine recommendations for safe dietary intakes. However, current data support the need for a paradigm shift that acknowledges that how we prepare and process food may be equally as important as nutrient composition.

However, increased dAGEs’ consumption, which is synonymous with the increased consumption of processed foods in terms of the “modern Western” diet, increases the total body AGEs’ burden. Most, but not all studies support that they are involved in the pathogenesis of various diseases and their complications, in a similar way and independently of those endogenously formed. These data may directly relate to humans, and dAGEs’ restriction seems to be a simple, safe alternative strategy for the prevention of various diseases or the delay of their progression. Ongoing studies are needed to further expand the dAGEs database and investigate additional methods for reducing AGEs’ generation during home cooking and food processing.

Future studies should continue to investigate the health effects of AGEs and refine recommendations for safe dietary intakes. However, current data support the need for a paradigm shift that acknowledges that how we prepare, and process food may be equally as important as nutrient composition.

## Figures and Tables

**Figure 1 nutrients-13-03467-f001:**
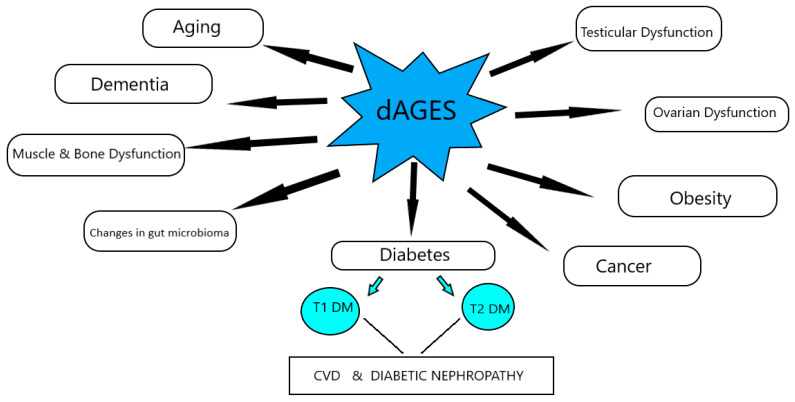
Effects of dAGEs on health and disease, according to animal studies.

**Table 1 nutrients-13-03467-t001:** Summary of dAGEs-related animal studies.

Author	Animal Model	Diet	Results
Thornton et al. 2020 [15]	C57BL/6 J mice	H-AGEs diet: 16.6% fat, 18.8% protein, 64.6% carbohydrate, 3.73 kcal/g, subjected to heating at 125 °C for 30 min	H-AGEs mice showed a lower number of corpora lutea
Rajan et al. 2018 [16]	Swiss albino mice	H-AGEs diet (18% of protein, 64% of carbohydrate, 5.2% of fat) subjected to heating (90 °C for 10 min) to generate diet-derived AGEs	Increased body weight, pro-inflammatory cytokines, chemokines, CML, CRP, HbA1c, BUN, creatinine
Deng et al. 2020 [17]	apoE−/− mice	High fat diet (42% fat and 0.2% cholesterol) and normal diet (13% fat, no cholesterol) and aminoguanidine	Elevation in the aortic valve transvalvular velocity, alleviation of LDL, TC reduced HFD-induced CML accumulation and RAGE expression in the aortic valve, and restricted EndMT in the aortic valve calcification
Mastrocola et al. 2021 [18]	C57BLKS-Db/Db mice	Standard diet and high fat diet (HFD) and pyridoxamine	Accumulation of AGEs in the liver impaired hepatic sphingolipids metabolism and hepatic insulin signaling, prevented by pyridoxamine
Verboven et al. 2018 [19]	Sprague–Dawley rats	High-sugar and high-fat diet (Western diet)	Altered glucose tolerance, early signs of cardiac alteration
Chen et al. 2019 [20]	Sprague–Dawley	L-AGEs and H-AGEs diets	H-AGEs diet decreases the total number of epididymal sperm and increases the abnormal sperm ratio
Mao et al. 2019 [21]	Sprague–Dawley rats	Low-fat control diet and high-fat diet	High fat diet increased cecal contents and decreased proteobacteria
Coughlan et al. 2011 [22]	Sprague–Dawley rats	L-AGEs diet and H-AGEs diet	H-AGEs diet increased cellular pathways linked to b-cell damage and b-cell apoptosis and the incidence of autoimmune diabetes in NOD mice
Coughlan et al. 2011 [22]	Sprague–Dawley rats	Daily intraperitoneal injections of either AGEs-modified rat serum albumin (AGE-RSA), RSA at 20 mg/kg/day, or saline (sham)	H-AGEs diet increased cellular pathways linked to b-cell damage and b-cell apoptosis and the incidence of autoimmune diabetes in NOD mice
Yuan et al. 2017 [23]	Sprague–Dawley rats	Intraperitoneal injection of high-fat/high-sucrose diet and low-dose streptozocin, untreated diabetic and treated with aminoguanidine hydrochloride (100 mg/Kg/day, i.g., for 8 weeks)	AG reduced serum and renal CML deposition, and improved urine protein and uNGAL in type 2 diabetic rats
Borg et al. 2018 [24]	NOD8.3 + NOD/ShiLt +NOD8.3C	H-AGEs diet and L-AGEs diet	Exposure to low dietary AGEs from conception to early postnatal life increased islet hormone secretion ex vivo; reduced insulitis; increased the variance of CD4C and CD8C T cells and cDCs in local lymphoid tissues and proportions of pDCs in the spleen; and altered islet expression of the AGEs, CML, and AGE receptors
Lee et al. 2019 [25]	Sprague–Dawley rats	Normal diet and regular, low, or high high-fat high-sucrose diet	L-AGEs showed lower fasting glucose, insulin, HOMA-IR, TC, TG, HDLc levels, higher blood urea nitrogen, n-3 PUFAs, decreased proliferation of the mesangial cells, glomerular capillaries, basement membranes around the glomeruli, lower Firmicutes/Bacteroidetes ratio, and increased Allobaculum
Maciejczyk et al. 2018 [26]	male Wistar rats	High fat diet (59.8 kcal% fat, 20.1 kcal% protein, and 20.1 kcal% carbs) and normal diet (g 13.5 kcal% fat, 24 kcal% protein, and 62.5 kcal% carbohydrates)	High fat fed rats showed increased body weight, free fatty acids, glucose, insulin, HOMA-IR index, enzymatic antioxidant activity, total antioxidant/oxidant status and oxidative damage products (TAC, TOS, OSI, and FRAP), activity of NOX and XO) in the cerebral cortex and hypothalamus, and enzymatic antioxidants (GPx, CAT, and SOD-1) in the cerebral cortex
Qu et al. 2017 [27]	Sprague–Dawley rats	L-AGEs diet or high-AGEs diet	H-AGEs fed rats showed markedly decreased diversity of cecal microbiota, after 18 weeks of feeding, increased proportion of proteobacteria, and decreased proportion of Bacteroidetes; in the short-term feeding period of 6 weeks, significantly higher relative abundance of five genera, including Prevotella, Oscillibacter, Phascolarctobacterium, Akkermansia, and Gastranaerophilales; higher concentration of ammonia in cecal contents and significantly lower concentration of two other genera, Lachnospiraceae and Mucispirillum at 12 and 18 weeks of feeding; decreased acetate and propionate from 6 to 18 weeks; and modestly increased butyric acid and histological score of colonic tissue
Illien-Juünger et al. 2018 [28]	C57BL/6J mice	L-AGEs; containing 7.6 μg/mg AGE and H-AGEs: 40.9 μg/m	H-AGEs fed mice showed elevated serum AGEs levels in female mice, sex- and age-dependent effects on vertebral AGEs accumulation and on vertebral bone microstructure, and decreased vertebral mechanical properties
Peppa et al. 2003 [29]	Prediabetic NOD mice	H-AGEs diet produced vy exposure to heating (100 °C for 20–60 s and at 125 °C for 20–30 min) and L-AGEs = identical chow mix without heating	L-AGEs fed mice showed a striking reduction in fasting blood glucose, increased plasma insulin levels, decreased affected (20%) pancreatic islets, low levels of IFN-γ and IL-4, high IL-10-to-actin mRNA ratio, and prevention of Type 1 diabetes transmitted to next generations
Peppa et al. 2003 [30]	db/db +/+ and db/db −/+	H-AGEs diet produced by heating (100 °C for 20–60 min and at 125 °C for 20–30 min), L-AGEs identical chow mix without heating	H-AGEs decreased the albumin/creatinine ratio, increased protein-linked tissue deposition of MG- and CML-like AGEs, delayed closure, lead to less re-epithelialization, and delayed wound healing
Vlassara et al. 1992 [31]	Lewis rats and New Zealand White rabbits	Tail vein injections with either AGEs-modified or native RSA (100 mg/kg per day) or AGEs-RSA, followed immediately by i.v. injection of aminoguanidine hydrochloride (100 mg/kg per day)	AGEs’ administration resulted in significantly increased vascular permeability, mononuclear cell migratory activity in subendothelial and periarteriolar spaces in various tissues, markedly defective vasodilatory responses to acetylcholine and nitroglycerin, promoted glomerulosclerosis, and normal characteristics in treated with aminoguanidine
Cai et al. 2007 [32]	C57BL/6 mice	H-AGEs diet and L-AGEs diet	L-AGEs fed mice showed amelioration of insulin resistance, albuminuria, and glomerulosclerosis, as well as extended lifespan
Hofmann et al. 2002 [33]	db/db (+/+)	H-AGEs diet and L-AGEs diet	L-AGEs fed mice showed reduced body weight, improved responses to both glucose and insulin tolerance tests, increased HDL and lowered CML and MG concentrations, and better preservation of the islets
Sandu et al. 2005 [34]	C57/BL6 mice	Regular and high-fat diets	H-AGEs fed mice showed higher body weight, fasting glucose, insulin, serum AGEs, altered pancreatic islet structure and function, plasma 8-isoprostanes, and lower adiponectin
Cai et al. 2012 [35]	wild-type C57BL6 mice	Nonheated, isocaloric diet, where the content of AGEs was increased by a single synthetic MG-AGE (MG+)	High fat fed mice showed increased adiposity and premature insulin resistance; severe deficiency of AGER1 and SIRT1 in white adipose tissue, skeletal muscle, and liver; impaired 2-deoxy-glucose uptake; marked changes in insulin receptor IRS-1, IRS-2; Akt activation; and a macrophage and adipocyte shift to a pro-oxidant/inflammatory phenotype
Cai et al. 2008 [36]	C57BL/6 mice	Caloric restriction (40% reduction in calories), caloric restriction exposed to heating, the same CR diet, in which the content of (by 15 min at 120 °C, and standard formula)	CR-high AGEs fed mice showed high levels of 8-isoprostanes, AGEs, RAGE, p66shc, low AGER1 and GSH/GSSG levels, insulin resistance, marked myocardial and renal fibrosis, and shortened lifespan
Kandaraki et al. 2012 [37]	Wistar rats	L-AGEs diet and H-AGEs diet	H-AGEs fed rats showed reduced GLO-I activity, positively correlated with body weight gain and progesterone levels
Zheng et al. 2002 [38]	db/db mice	L-AGEs diet and H-AGEs diet	L-AGEs fed mice showed minimal glomerular pathology; modest increase in urinary albumin/creatinine ratio; extended survival; lower serum; and kidney AGEs low levels of renal cortex TGFb-1, laminin B1 mRNA, a1 IV collagen mRNA, and protein

Abbreviations: L-AGEs diet = low in advanced glycation end products diet; H-AGEs diet = high in advanced glycation end products diet; HFD = high fat diet; CR = caloric reduction; HF = high fat; CML = Nε-carboxymethyllysine; MG = methylglyoxal; AG = aminoguanidine (AGE-inhibitor); CRP = c reactive protein; HbA1c = hemoglobin A1c; BUN = blood urea nitrogen; LDL = low density lipoprotein; TG = triglycerides; RAGE = advanced glycation end products receptor; EndMT = endothelial to mesenchymal transition; HOMA-IR; TC = tissue culture; HDL = high density lipoprotein; n-3 PUFAs = n-3 polyunsaturated fatty acids; TAC = total antioxidant; TOS = total oxidant status; FRAP = ferric reducing ability; NOX = catalytic subunit of nicotinamide adenine dinucleotide phosphate (NADPH) oxidases; GPx = glutathione peroxidase; CAT = catalase; SOD-1 = Cu, Zn-superoxide dismutase-1; IFN-γ = interferon-γ; IL-4 = interleukin-4; IL-10 = interleukin-10; AGER1 = advanced glycation end product receptor-1; SIRT1 = sirtuin-1; receptor IRS-1 = insulin receptor 1; IRS-2 = insulin receptor 2; Akt = serine/threonine kinase 1; GSH = glutathione, GSSG = glutamic acid, cysteine, and glycine glutathione; TGFb-1 = transforming growth factor beta-1.

## Data Availability

Not applicable.

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
