# Peer review of "Experimental Animal Studies Support the Role of Dietary Advanced Glycation End Products in Health and Disease"

_nutrients, 2021, doi:10.3390/nu13103467_

Round 1

Reviewer 1 Report

in the manuscript authored by Melpomeni Peppa et al. authors provided an interesting review about the role of dietary AGEs in health and disease. For this purpose, authors decided to revise the literature focalizing their attention mainly on animal model studies. 

Here below I reported my comments and suggestions:

-Several typos or punctuation mistakes are present throughout the entire manuscript. see "indi-cates" or T1D??? 

-Figure 1 should be improved in image quality.

-Authors refer to AGER1 or AGE receptor 1. please uniform the style.

-Authors usually report H-AGEs and L-AGEs in all the studies reported. Despite I wonder if there are commonly shared and standardized H-AGEs and L-AGEs diets in literature? when a diet can be considered low or high? there is a respective threshold that can be transposed to humans?

I suggest author also include these more details in the manuscript. 

-I also suggest authors discuss with a critical point of view limits and criticism of animal studies considering also their translation to humans.

-A summary table should be included to better highlight the evidence of low AGEs to promote health that should include mice models and the diet used in the study. 

Author Response

First I would like to thank you for the review and I hope I responded adequately in your comments.  

In the manuscript authored by Melpomeni Peppa et al. authors provided an interesting review about the role of dietary AGEs in health and disease. For this purpose, authors decided to revise the literature focalizing their attention mainly on animal model studies. 

Here below I reported my comments and suggestions:

-Several typos or punctuation mistakes are present throughout the entire manuscript. see "indi-cates" or T1D??? 

We looked carefully throughout the manuscript and corrected them

-Figure 1 should be improved in image quality.

We improved the image quality  

-Authors refer to AGER1 or AGE receptor 1. please uniform the style.

AGER-1 is the abbreviation of AGE receptor 1 and we made the changes where they needed ( page 5 lines 174,203,204, page 7 lines 289,298,310)

-Authors usually report H-AGEs and L-AGEs in all the studies reported. Despite I wonder if there are commonly shared and standardized H-AGEs and L-AGEs diets in literature? when a diet can be considered low or high? there is a respective threshold that can be transposed to humans?

A new paragraph has been inserted in pages 2 and 3, entitled “Dietary AGEs Homeostasis—Animal Studies”, in which we tried to make the requested points

-I also suggest authors discuss with a critical point of view limits and criticism of animal studies considering also their translation to humans.

A paragraph has been inserted in pages 10 and 11, entitled  “Limitations of animal studies in dAGEs”, in which we tried to make the requested points

-A summary table should be included to better highlight the evidence of low AGEs to promote health that should include mice models and the diet used in the study. 

A summary table has been created and inserted in the paper , but it was impossible to upload

Reviewer 2 Report

The authors present a review on animal data supporting the role of dietary advanced glycation end products (AGE) in the pathogenesis of several conditions such as diabetes mellitus, diabetic vasculopathy, diabetic nephropathy, obesity, aging, dementia, gut microbiome disturbances, and reproductive function impairment. The issue is of broad interest because AGE rich western diet is most likely causing increasing numbers of AGE associated morbidity and mortality. The paper is clearly structured and easy to read. I have only minor concerns: The authors should include a brief section on the definition of AGE and the characteristics of low and high AGE diet. The authors should also comment on the association of AGE, known cancer risk factors, and cancer. Some question marks in the text must be removed. Some references in the text (8, 16, 43, and 44) apparently do not fit with the references list in the end. Please explain the acronym CFA in line 345 on page 8. The selection of refences and the author’s conclusions are also very clear. However, the authors should include a brief statement whether conflicting studies are available or not. I cannot imagine that all animal studies came to the same results/conclusions.

Author Response

REVIEWER 2

The authors present a review on animal data supporting the role of dietary advanced glycation end products (AGE) in the pathogenesis of several conditions such as diabetes mellitus, diabetic vasculopathy, diabetic nephropathy, obesity, aging, dementia, gut microbiome disturbances, and reproductive function impairment. The issue is of broad interest because AGE rich western diet is most likely causing increasing numbers of AGE associated morbidity and mortality. The paper is clearly structured and easy to read. I have only minor concerns:

The authors should include a brief section on the definition of AGE and the characteristics of low and high AGE diet.

A new paragraph has been inserted in pages 2 and 3, entitled “Dietary AGEs Homeostasis—Animal Studies”, in which we tried to make the requested points

-The authors should also comment on the association of AGE, known cancer risk factors, and cancer.

We introduced a new paragraph entitled “Dietary AGEs effects on cancer” , in page 10

-Some question marks in the text must be removed.

We removed the question marks

Some references in the text (8, 16, 43, and 44) apparently do not fit with the references list in the end.

We edited the references

-Please explain the acronym CFA in line 345 on page 8.

CFA is the abbreviation of small chain fatty acids, and was corrected

-The selection of references and the author’s conclusions are also very clear. However, the authors should include a brief statement whether conflicting studies are available or not. I cannot imagine that all animal studies came to the same results/conclusions. 

A statement about conflicting results have been made in the paragraph “limitations of animal studies in dAGEs”

REVIEWER 2

The authors present a review on animal data supporting the role of dietary advanced glycation end products (AGE) in the pathogenesis of several conditions such as diabetes mellitus, diabetic vasculopathy, diabetic nephropathy, obesity, aging, dementia, gut microbiome disturbances, and reproductive function impairment. The issue is of broad interest because AGE rich western diet is most likely causing increasing numbers of AGE associated morbidity and mortality. The paper is clearly structured and easy to read. I have only minor concerns:

The authors should include a brief section on the definition of AGE and the characteristics of low and high AGE diet.

A new paragraph has been inserted in pages 2 and 3, entitled “Dietary AGEs Homeostasis—Animal Studies”, in which we tried to make the requested points

-The authors should also comment on the association of AGE, known cancer risk factors, and cancer.

We introduced a new paragraph entitled “Dietary AGEs effects on cancer” , in page 10

-Some question marks in the text must be removed.

We removed the question marks

Some references in the text (8, 16, 43, and 44) apparently do not fit with the references list in the end.

We edited the references

-Please explain the acronym CFA in line 345 on page 8.

CFA is the abbreviation of small chain fatty acids, and was corrected

-The selection of references and the author’s conclusions are also very clear. However, the authors should include a brief statement whether conflicting studies are available or not. I cannot imagine that all animal studies came to the same results/conclusions. 

A statement about conflicting results have been made in the paragraph “limitations of animal studies in dAGEs”

REVIEWER 2

The authors present a review on animal data supporting the role of dietary advanced glycation end products (AGE) in the pathogenesis of several conditions such as diabetes mellitus, diabetic vasculopathy, diabetic nephropathy, obesity, aging, dementia, gut microbiome disturbances, and reproductive function impairment. The issue is of broad interest because AGE rich western diet is most likely causing increasing numbers of AGE associated morbidity and mortality. The paper is clearly structured and easy to read. I have only minor concerns:

The authors should include a brief section on the definition of AGE and the characteristics of low and high AGE diet.

A new paragraph has been inserted in pages 2 and 3, entitled “Dietary AGEs Homeostasis—Animal Studies”, in which we tried to make the requested points

-The authors should also comment on the association of AGE, known cancer risk factors, and cancer.

We introduced a new paragraph entitled “Dietary AGEs effects on cancer” , in page 10

-Some question marks in the text must be removed.

We removed the question marks

Some references in the text (8, 16, 43, and 44) apparently do not fit with the references list in the end.

We edited the references

-Please explain the acronym CFA in line 345 on page 8.

CFA is the abbreviation of small chain fatty acids, and was corrected

-The selection of references and the author’s conclusions are also very clear. However, the authors should include a brief statement whether conflicting studies are available or not. I cannot imagine that all animal studies came to the same results/conclusions. 

A statement about conflicting results have been made in the paragraph “limitations of animal studies in dAGEs”

REVIEWER 2

The authors present a review on animal data supporting the role of dietary advanced glycation end products (AGE) in the pathogenesis of several conditions such as diabetes mellitus, diabetic vasculopathy, diabetic nephropathy, obesity, aging, dementia, gut microbiome disturbances, and reproductive function impairment. The issue is of broad interest because AGE rich western diet is most likely causing increasing numbers of AGE associated morbidity and mortality. The paper is clearly structured and easy to read. I have only minor concerns:

The authors should include a brief section on the definition of AGE and the characteristics of low and high AGE diet.

A new paragraph has been inserted in pages 2 and 3, entitled “Dietary AGEs Homeostasis—Animal Studies”, in which we tried to make the requested points

-The authors should also comment on the association of AGE, known cancer risk factors, and cancer.

We introduced a new paragraph entitled “Dietary AGEs effects on cancer” , in page 10

-Some question marks in the text must be removed.

We removed the question marks

Some references in the text (8, 16, 43, and 44) apparently do not fit with the references list in the end.

We edited the references

-Please explain the acronym CFA in line 345 on page 8.

CFA is the abbreviation of small chain fatty acids, and was corrected

-The selection of references and the author’s conclusions are also very clear. However, the authors should include a brief statement whether conflicting studies are available or not. I cannot imagine that all animal studies came to the same results/conclusions. 

A statement about conflicting results have been made in the paragraph “limitations of animal studies in dAGEs”

REVIEWER 2

The authors present a review on animal data supporting the role of dietary advanced glycation end products (AGE) in the pathogenesis of several conditions such as diabetes mellitus, diabetic vasculopathy, diabetic nephropathy, obesity, aging, dementia, gut microbiome disturbances, and reproductive function impairment. The issue is of broad interest because AGE rich western diet is most likely causing increasing numbers of AGE associated morbidity and mortality. The paper is clearly structured and easy to read. I have only minor concerns:

The authors should include a brief section on the definition of AGE and the characteristics of low and high AGE diet.

A new paragraph has been inserted in pages 2 and 3, entitled “Dietary AGEs Homeostasis—Animal Studies”, in which we tried to make the requested points

-The authors should also comment on the association of AGE, known cancer risk factors, and cancer.

We introduced a new paragraph entitled “Dietary AGEs effects on cancer” , in page 10

-Some question marks in the text must be removed.

We removed the question marks

Some references in the text (8, 16, 43, and 44) apparently do not fit with the references list in the end.

We edited the references

-Please explain the acronym CFA in line 345 on page 8.

CFA is the abbreviation of small chain fatty acids, and was corrected

-The selection of references and the author’s conclusions are also very clear. However, the authors should include a brief statement whether conflicting studies are available or not. I cannot imagine that all animal studies came to the same results/conclusions. 

A statement about conflicting results have been made in the paragraph “limitations of animal studies in dAGEs”

REVIEWER 2

The authors present a review on animal data supporting the role of dietary advanced glycation end products (AGE) in the pathogenesis of several conditions such as diabetes mellitus, diabetic vasculopathy, diabetic nephropathy, obesity, aging, dementia, gut microbiome disturbances, and reproductive function impairment. The issue is of broad interest because AGE rich western diet is most likely causing increasing numbers of AGE associated morbidity and mortality. The paper is clearly structured and easy to read. I have only minor concerns:

The authors should include a brief section on the definition of AGE and the characteristics of low and high AGE diet.

A new paragraph has been inserted in pages 2 and 3, entitled “Dietary AGEs Homeostasis—Animal Studies”, in which we tried to make the requested points

-The authors should also comment on the association of AGE, known cancer risk factors, and cancer.

We introduced a new paragraph entitled “Dietary AGEs effects on cancer” , in page 10

-Some question marks in the text must be removed.

We removed the question marks

Some references in the text (8, 16, 43, and 44) apparently do not fit with the references list in the end.

We edited the references

-Please explain the acronym CFA in line 345 on page 8.

CFA is the abbreviation of small chain fatty acids, and was corrected

-The selection of references and the author’s conclusions are also very clear. However, the authors should include a brief statement whether conflicting studies are available or not. I cannot imagine that all animal studies came to the same results/conclusions. 

A statement about conflicting results have been made in the paragraph “limitations of animal studies in dAGEs”

REVIEWER 2

The authors present a review on animal data supporting the role of dietary advanced glycation end products (AGE) in the pathogenesis of several conditions such as diabetes mellitus, diabetic vasculopathy, diabetic nephropathy, obesity, aging, dementia, gut microbiome disturbances, and reproductive function impairment. The issue is of broad interest because AGE rich western diet is most likely causing increasing numbers of AGE associated morbidity and mortality. The paper is clearly structured and easy to read. I have only minor concerns:

The authors should include a brief section on the definition of AGE and the characteristics of low and high AGE diet.

A new paragraph has been inserted in pages 2 and 3, entitled “Dietary AGEs Homeostasis—Animal Studies”, in which we tried to make the requested points

-The authors should also comment on the association of AGE, known cancer risk factors, and cancer.

We introduced a new paragraph entitled “Dietary AGEs effects on cancer” , in page 10

-Some question marks in the text must be removed.

We removed the question marks

Some references in the text (8, 16, 43, and 44) apparently do not fit with the references list in the end.

We edited the references

-Please explain the acronym CFA in line 345 on page 8.

CFA is the abbreviation of small chain fatty acids, and was corrected

-The selection of references and the author’s conclusions are also very clear. However, the authors should include a brief statement whether conflicting studies are available or not. I cannot imagine that all animal studies came to the same results/conclusions. 

A statement about conflicting results have been made in the paragraph “limitations of animal studies in dAGEs”

REVIEWER 2

The authors present a review on animal data supporting the role of dietary advanced glycation end products (AGE) in the pathogenesis of several conditions such as diabetes mellitus, diabetic vasculopathy, diabetic nephropathy, obesity, aging, dementia, gut microbiome disturbances, and reproductive function impairment. The issue is of broad interest because AGE rich western diet is most likely causing increasing numbers of AGE associated morbidity and mortality. The paper is clearly structured and easy to read. I have only minor concerns:

The authors should include a brief section on the definition of AGE and the characteristics of low and high AGE diet.

A new paragraph has been inserted in pages 2 and 3, entitled “Dietary AGEs Homeostasis—Animal Studies”, in which we tried to make the requested points

-The authors should also comment on the association of AGE, known cancer risk factors, and cancer.

We introduced a new paragraph entitled “Dietary AGEs effects on cancer” , in page 10

-Some question marks in the text must be removed.

We removed the question marks

Some references in the text (8, 16, 43, and 44) apparently do not fit with the references list in the end.

We edited the references

-Please explain the acronym CFA in line 345 on page 8.

CFA is the abbreviation of small chain fatty acids, and was corrected

-The selection of references and the author’s conclusions are also very clear. However, the authors should include a brief statement whether conflicting studies are available or not. I cannot imagine that all animal studies came to the same results/conclusions. 

A statement about conflicting results have been made in the paragraph “limitations of animal studies in dAGEs”

REVIEWER 2

The authors present a review on animal data supporting the role of dietary advanced glycation end products (AGE) in the pathogenesis of several conditions such as diabetes mellitus, diabetic vasculopathy, diabetic nephropathy, obesity, aging, dementia, gut microbiome disturbances, and reproductive function impairment. The issue is of broad interest because AGE rich western diet is most likely causing increasing numbers of AGE associated morbidity and mortality. The paper is clearly structured and easy to read. I have only minor concerns:

The authors should include a brief section on the definition of AGE and the characteristics of low and high AGE diet.

A new paragraph has been inserted in pages 2 and 3, entitled “Dietary AGEs Homeostasis—Animal Studies”, in which we tried to make the requested points

-The authors should also comment on the association of AGE, known cancer risk factors, and cancer.

We introduced a new paragraph entitled “Dietary AGEs effects on cancer” , in page 10

-Some question marks in the text must be removed.

We removed the question marks

Some references in the text (8, 16, 43, and 44) apparently do not fit with the references list in the end.

We edited the references

-Please explain the acronym CFA in line 345 on page 8.

CFA is the abbreviation of small chain fatty acids, and was corrected

-The selection of references and the author’s conclusions are also very clear. However, the authors should include a brief statement whether conflicting studies are available or not. I cannot imagine that all animal studies came to the same results/conclusions. 

A statement about conflicting results have been made in the paragraph “limitations of animal studies in dAGEs”

REVIEWER 2

The authors present a review on animal data supporting the role of dietary advanced glycation end products (AGE) in the pathogenesis of several conditions such as diabetes mellitus, diabetic vasculopathy, diabetic nephropathy, obesity, aging, dementia, gut microbiome disturbances, and reproductive function impairment. The issue is of broad interest because AGE rich western diet is most likely causing increasing numbers of AGE associated morbidity and mortality. The paper is clearly structured and easy to read. I have only minor concerns:

The authors should include a brief section on the definition of AGE and the characteristics of low and high AGE diet.

A new paragraph has been inserted in pages 2 and 3, entitled “Dietary AGEs Homeostasis—Animal Studies”, in which we tried to make the requested points

-The authors should also comment on the association of AGE, known cancer risk factors, and cancer.

We introduced a new paragraph entitled “Dietary AGEs effects on cancer” , in page 10

-Some question marks in the text must be removed.

We removed the question marks

Some references in the text (8, 16, 43, and 44) apparently do not fit with the references list in the end.

We edited the references

-Please explain the acronym CFA in line 345 on page 8.

CFA is the abbreviation of small chain fatty acids, and was corrected

-The selection of references and the author’s conclusions are also very clear. However, the authors should include a brief statement whether conflicting studies are available or not. I cannot imagine that all animal studies came to the same results/conclusions. 

A statement about conflicting results have been made in the paragraph “limitations of animal studies in dAGEs”

REVIEWER 2

The authors present a review on animal data supporting the role of dietary advanced glycation end products (AGE) in the pathogenesis of several conditions such as diabetes mellitus, diabetic vasculopathy, diabetic nephropathy, obesity, aging, dementia, gut microbiome disturbances, and reproductive function impairment. The issue is of broad interest because AGE rich western diet is most likely causing increasing numbers of AGE associated morbidity and mortality. The paper is clearly structured and easy to read. I have only minor concerns:

The authors should include a brief section on the definition of AGE and the characteristics of low and high AGE diet.

A new paragraph has been inserted in pages 2 and 3, entitled “Dietary AGEs Homeostasis—Animal Studies”, in which we tried to make the requested points

-The authors should also comment on the association of AGE, known cancer risk factors, and cancer.

We introduced a new paragraph entitled “Dietary AGEs effects on cancer” , in page 10

-Some question marks in the text must be removed.

We removed the question marks

Some references in the text (8, 16, 43, and 44) apparently do not fit with the references list in the end.

We edited the references

-Please explain the acronym CFA in line 345 on page 8.

CFA is the abbreviation of small chain fatty acids, and was corrected

-The selection of references and the author’s conclusions are also very clear. However, the authors should include a brief statement whether conflicting studies are available or not. I cannot imagine that all animal studies came to the same results/conclusions. 

A statement about conflicting results have been made in the paragraph “limitations of animal studies in dAGEs”

REVIEWER 2

The authors present a review on animal data supporting the role of dietary advanced glycation end products (AGE) in the pathogenesis of several conditions such as diabetes mellitus, diabetic vasculopathy, diabetic nephropathy, obesity, aging, dementia, gut microbiome disturbances, and reproductive function impairment. The issue is of broad interest because AGE rich western diet is most likely causing increasing numbers of AGE associated morbidity and mortality. The paper is clearly structured and easy to read. I have only minor concerns:

The authors should include a brief section on the definition of AGE and the characteristics of low and high AGE diet.

A new paragraph has been inserted in pages 2 and 3, entitled “Dietary AGEs Homeostasis—Animal Studies”, in which we tried to make the requested points

-The authors should also comment on the association of AGE, known cancer risk factors, and cancer.

We introduced a new paragraph entitled “Dietary AGEs effects on cancer” , in page 10

-Some question marks in the text must be removed.

We removed the question marks

Some references in the text (8, 16, 43, and 44) apparently do not fit with the references list in the end.

We edited the references

-Please explain the acronym CFA in line 345 on page 8.

CFA is the abbreviation of small chain fatty acids, and was corrected

-The selection of references and the author’s conclusions are also very clear. However, the authors should include a brief statement whether conflicting studies are available or not. I cannot imagine that all animal studies came to the same results/conclusions. 

A statement about conflicting results have been made in the paragraph “limitations of animal studies in dAGEs”
